# OpenReview forum: "Benchmarking Agent Memory in Interdependent Multi-Session Agentic Tasks"
_ICML.cc/2026/Conference — ICML 2026 regular_

### Official Review · Reviewer_ZqnJ · 2026-03-12

**Soundness:** 2
**Presentation:** 3
**Significance:** 3
**Originality:** 3
**Overall Recommendation:** 5
**Confidence:** 4

**Summary:**

This paper introduces MEMORYARENA, a benchmark for evaluating long-term memory in LLM agents under realistic multi-session interaction settings. In contrast to prior work that mainly focuses on static recall tasks (e.g., QA or summarization) or single-session environments, MEMORYARENA emphasizes closed-loop interaction between memory, agent decisions, and environment feedback across sessions. The benchmark covers four scenarios: bundled shopping, group travel planning, progressive search, and formal reasoning, with explicitly designed cross-session dependencies between subtasks. Through experiments on long-context models, retrieval-based systems, and specialized memory agents such as MemGPT, the paper shows that strong performance on static memory benchmarks does not transfer to successful decision-making in interactive environments. Performance drops substantially as dependency depth increases, suggesting that existing memory mechanisms struggle to preserve and use long-range constraints. The results also indicate that, in some cases, simple long-context prompting can outperform more complex external memory architectures.

**Compliance With Llm Reviewing Policy:**

Affirmed.

**Final Justification:**

My concerns have been addressed.

**Key Questions For Authors:**

1. Where is the primary bottleneck in the memory pipeline?

The benchmark reveals large performance drops for existing memory architectures, but it is unclear whether the main difficulty lies in information acquisition, memory encoding, retrieval, or downstream reasoning. A clearer decomposition of these stages would help determine whether the failures originate from memory mechanisms themselves or from the agent’s reasoning and planning components.

2. Is retrieval quality the dominant limiting factor?

In the current setup, the context window is reset between sessions, and the agent can only access prior information through the memory module. This suggests that the performance bottleneck may largely stem from retrieval quality rather than memory representation. It would be informative to evaluate stronger retrieval pipelines (e.g., multi-hop or deep-search-style retrieval agents) to test whether improved retrieval alone can significantly improve outcomes.

3. What happens under an oracle memory condition?

A useful diagnostic experiment would be to provide the agent with all and only the information required to infer the current belief state, effectively approximating a fully observable MDP. If performance improves substantially under such an oracle setting, this would indicate that the main challenge lies in information extraction and retrieval, rather than in reasoning or planning with memory.

4. Failure to remember vs. failure to observe.

The paper attributes poor performance primarily to limitations in memory mechanisms. However, it remains unclear whether the critical information was never obtained during earlier interactions, rather than being forgotten later. Quantifying how often key information is missed during earlier sessions versus lost in memory would clarify the true source of the failure.

5. Does error propagation confound the measurement of memory ability?

Because tasks have strong cross-session dependencies, early mistakes may propagate and invalidate later subtasks. As a result, later failures may not necessarily reflect memory limitations but rather the consequences of earlier incorrect actions. Evaluating later subtasks with oracle historical context could help isolate memory limitations from cascading decision errors.

**Limitations:**

The paper does not include a dedicated discussion of limitations. In particular, the manuscript does not address potential limitations such as the static nature of the dataset or possible evaluation errors introduced by automated judging.

**Strengths And Weaknesses:**

**Soundness**
- Strengths
1. The experimental results provide convincing evidence for the paper’s central observation that current memory mechanisms degrade significantly as task dependency depth increases.

2. The case studies in the appendix are particularly insightful. They clearly illustrate typical failure modes, such as retrieval failures in RAG systems and impulsive decision patterns in long-context models, which help readers understand the qualitative differences between memory architectures.

- Weaknesses
1. Hypotheses without empirical validation.

The paper proposes hypotheses such as representation mismatch and training mismatch between external memory systems and base language models. However, these claims remain largely speculative, as the paper does not provide supporting analyses (e.g., ablations, representation similarity analysis, or probing experiments) to empirically validate these mechanisms.

2. Unclear hyperparameter and prompting fairness across systems.

It is not entirely clear whether all systems were evaluated under sufficiently optimized settings. In particular, it would be useful to clarify whether systematic prompt engineering or task-specific prompt tuning was performed for complex reasoning tasks, and whether hyperparameters were carefully tuned for different memory architectures to ensure a fair comparison.

3. Limited dataset size in some scenarios.

Some domains appear relatively small (e.g., 270 instances for travel planning and only 20 for physical reasoning). It is unclear whether these sample sizes are sufficient to support strong conclusions about memory capabilities. With small evaluation sets, performance may also be sensitive to decoding randomness or sampling seeds.

4. Limited ablations on environment difficulty.

The paper categorizes memory systems into 0D/1D/2D architectures, but provides limited analysis of how specific environmental factors contribute to the difficulty. For example, it would be informative to evaluate a variant where the cross-session constraint is removed (i.e., a single long session) to quantify how much of the performance degradation is specifically caused by cross-session memory requirements.

5. Scope of the “unified evaluation gym.”

The benchmark is described as a unified evaluation gym, but the four scenarios still correspond to relatively specific vertical domains. It would be helpful to clarify in what sense the framework is “unified” and how well it generalizes beyond these particular environments.

6. Can opensource both MEMORYARENA infra and the corresponding dataset?

**Presentation**

The paper is generally well presented. The overall logical flow—from motivation to benchmark design and empirical evaluation—is clear and easy to follow, and the interaction loop between memory, agent actions, and environment feedback is illustrated effectively. In addition, the dataset construction and filtering pipeline is described transparently, including the two-stage filtering procedure using LLM-as-judge followed by human verification, which increases confidence in the quality of the benchmark.

**Significance**

The paper targets a key bottleneck for practical LLM agents, namely the gap between strong performance on static memory benchmarks and the ability to use memory effectively in realistic interactive settings. The proposed benchmark and environments could also serve as a reusable testbed for future research on long-term memory in agent systems.

**Originality**

The paper introduces the notion of explicit interdependency and provides the first systematic quantification of its impact in multi-session agent settings. However, the positioning with respect to closely related concurrent work (e.g., Evo-Memory) could be stronger; the paper does not clearly articulate a sharp conceptual or methodological boundary that distinguishes its core contribution from adjacent efforts.

---

> ### Author Rebuttal · Authors · 2026-03-31
>
> **W1:**  Our core contribution is the *benchmark*: new data revealing previously unobserved behaviors of memory systems in multi-session settings. We provide empirical findings with plausible explanations, rather than definitive causal claims (typically the focus of analysis papers), which are difficult given diverse memory designs.
>
> Encouragingly, Reviewer HuFN also found our mismatch explanation solid. Similar effects appear in prior work (e.g., SkillRL), where memory remains suboptimal without joint policy optimization, supporting our interpretation.
>
> **W2:**
> We use prompts adapted from prior work (WebShop, BrowseComp-Plus, Travel Planner) with a consistent setup. Memory systems follow original configurations with minimal changes. Outputs are manually verified to ensure correctness and fair comparison. We will release our codes, prompts, and hyperparameters.
>
> **W3:**
> MemoryArena is comparable in scale to agentic benchmarks (ToolAthlon:108, MCP-Bench:104, WebArena:821). Due to compositional multi-session design, it expands to **4,850 subtasks** (same task granularity as e.g. WebShop). Math/phys domains contain **440 subtasks**, yielding larger effective scale  and lower sensitivity to randomness.
>
> **W4:**
> A “single long session” variant is confounded by long-context degradation, making it hard to isolate cross-session memory effects from long-context limitations.  We agree this is an interesting direction and will also highlight it as future work.
>
> **W5:**
> *Unified* refers to the framework: MemoryArena provides a consistent pipeline where memory is a modular component that can be integrated with different agents and environments under a shared evaluation protocol.
>
> **coverage:** We include 4 representative settings. While not exhaustive, this provides an extensible foundation for future domains. Current MemoryArena is developed with substantial effort (~$60K), making it a high-quality starting point.
>
> **W6:**
> We will fully release the dataset and infra codebase.
>
> **Originality:**
> Our core contribution is establishing a new standardization for agentic memory using interdependent, multi-session tasks. This is fundamentally different from prior work and cannot reuse any existing datasets. Such data creation is non-trivial and resource-intensive (e.g., ~10h per math/physics task by senior PhD annotators). We will further emphasize our contribution in the revision.
>
> **Q1–Q3 (Unified clarification)**
>
> **A. Non-decomposability:** Memory in multi-session agents is inherently compositional. However, memory systems are heterogeneous and often black-box, making a unified stage-wise decomposition or an “oracle” intervention ill-defined. Instead, MemoryArena evaluates memory end-to-end, where success causally depends on correct memory usage, complementing prior stage-isolated (e.g., recall-based) benchmarks.
>
> **B. Retrieval is not dominant:** Long-context agents (no information loss, no retrieval) still perform poorly. Moreover, systems with strong retrieval (e.g., GraphRAG has a multi-hop retriever, Letta uses deep-search-style agents) also struggle. All indicates failures arise from interactions between memory construction, representation, and reasoning, not retrieval alone.
>
> **Q1**: See above #A.
>
> **Q2**:  While retrieval matters for classical RAG (Table3: changing BM25 retrieval to dense vector retrieval can improve the outcome), it is not dominant in agentic memory systems (#B).
>
> **Q3**:  An oracle memory is difficult to define (#A). Importantly, #B already provides a strong proxy: even with perfect information access (long-context), performance remains low, indicating additional challenges beyond information extraction and retrieval.
>
> **Q4:**  We conduct an additional analysis by grouping constraints by their causal dependency distance (L0–L4). Pass rates decrease monotonically with depth and eventually drop to zero. Crucially, Non-zero performance at shallow levels shows information from earlier sessions is acquired, while degradation indicates failure to retain/use it later.
> Moreover, even long-context agents (all prior information is preserved) still perform poorly (Table3). Together, these results indicate that the primary issue is not failure to observe.
>
> **Q5:**
> Though interdependency is a realistic property of agentic env, error propagation is indeed inherent. We conduct a diagnostic by injecting ground-truth subtask outcomes into memory (removing dependence on prior errors), which yields no significant improvement (e.g., Shopping w/ Mirix: 0.008 diff), indicating that cascading errors alone do not explain the poor performance. Moreover, methods designed for independent tasks (e.g., procedural memory ReasoningBank, no error propagation) also struggle, showing fundamental challenges in memory utilization beyond cascading errors.
>
> Thanks for the insightful questions. We will add the discussions in the final version.
>
> **L1**: Thanks for the suggestion, we will add a limitation section accordingly.

---

> > ### Author Rebuttal · Reviewer_ZqnJ · 2026-04-03
> >
> > 1. In terms of W3, I understand that the total number of subtasks is sufficient for the benchmark. However, my main question is related to the underrepresented field, such as math or physics (as shown in Table 2, only 40 and 20 samples for each). But, you claimed that '440 subtasks' in total, so is there any misunderstanding? I think it's better to open-source for more clarity.
> >
> > 2. For W4, as you realised that the cross-session memory limitation is confounded with long-context degradation, I wish to see how much contribution is from the long-context problem, as we know your current benchmark measures everything as a whole. Then, we can better understand the underlying mechanism of the cross-session memory limit.
> >
> > 3. For Oracle access, what I am concerned about is the key context required to fulfil the final task. For example, I would like to know if we give all key information for LLM, then can it improve substantially compared to the current poor performance on your benchmark? Also, on the other side, for retrieval, I mean the general information seeking process, which means I would like to know the challenges that happen in information seeking or how LLMs utilise them.
> >
> > 4. I understand 'Non-zero performance at shallow levels shows information from earlier sessions is acquired', but the key question is that your sample is cascaded, which means with more chain, the problem of failure to observe will be more severe, so I wish to see more details for the impact on the later levels.
> >
> > 5. 'no significant improvement (e.g., Shopping w/ Mirix: 0.008 diff), indicating that cascading errors alone do not explain the poor performance' for this one, can you show me the full results table of comparison?  for your four tasks.

---

> > > ### Author Response · Authors · 2026-04-04
> > >
> > > Thank you, we are happy to continue discussion.
> > > ## Clarification: subtask
> > > We believe there is a misunderstanding between tasks and subtasks. In MemoryArena, each environment has multiple tasks, and each task is decomposed into multiple interdependent subtasks.
> > >
> > > Thus, the reported 440 subtasks refers to the total number of flattened subtasks across math/phys tasks (354 Math, 86 Phys; comparable to TheoremQA: 442/131), not top-level tasks. **Each subtask corresponds to a lemma/proposition/theorem proof at a granularity comparable to prior benchmarks**, providing meaningful scale and reducing randomness.
> > >
> > > Below is an example of a math subtask from [1] that solves a *information upper bound* to support later subtasks.
> > > > (Proposition 3) Let $Y \in \\{0,1\\}^{m, b}$ be the transcript generated by a distributed estimation protocol for $k$-TPCA with parameters $(m, 1, b)$. Then what is the upper bound the mutual information $\textbf{I}_\text{hel}({\textbf{V};\textbf{Y}})$, where
> > > $\\sigma^2 {:=}  \begin{cases} C_k \cdot \lambda^2 \cdot  d^{-\frac{k+2}{2}} & \text{if $k$ is even} , \\  C_k \cdot \lambda^2 \cdot  d^{-\frac{k+1}{2}} & \text{if $k$ is odd} ; \end{cases}$  and $C_k>0$ is a positive constant that depends only on $k$. (background&definitions omitted).
> > >
> > > *Answer:*
> > > > $\left( \sigma^2 \cdot m \cdot b + \frac{1}{d}  + \lambda^2  \cdot b \cdot  \left( \frac{\lambda^2  \vee \ln(m \cdot d}{d}\right)^{\frac{k}{2}} + \inf_{\alpha \geq 2} \frac{\lambda^2 \alpha}{d} + m \cdot \left(\frac{C_k \alpha \lambda^2}{\sqrt{d^k}} + e^{-d} \right)^{\frac{\alpha}{2}}  \right)$
> > >
> > > *This subtask spans ~8 pages of derivation in [1] (pages 55-63).*
> > >
> > > As illustrated, **our subtasks are at research-paper-level complexity and significantly harder than benchmarks like GSM8K or TheoremQA** (and thus provide meaningful scale). Thus, we view our math/phys data as high-quality complementary resources.
> > >
> > > **Open-source:** We promise to fully release all data and code.
> > >
> > > [1] arXiv:2204.07526
> > > ## Long-context contribution
> > > We agree this is an interesting direction. However, cleanly disentangling long-context degradation from cross-session dependency is inherently difficult, as they jointly affect performance.
> > >
> > > However, we try our best to design a setting with reduced long-context degradation by distilling previous long reasoning traces into short step-by-step summaries, while also injecting golden outcomes from previous subtasks to break error dependency from previous sessions. We evaluate this using long-context agents so that there is no retrieval loss.
> > > |  | Shopping | Travel | Search | Math |
> > > |---|---|---|---|---|
> > > | $\Delta$ | +0.016 | +0.050 | +0.54 | +0.16 |
> > >
> > > We observe consistent improvements across environments (especially for search tasks where the original traces are overly long). While this intervention also introduces additional effects (e.g., knowledge condensation/abstraction), it provides indicative evidence of the impact of long-context degradation.
> > > ## Oracle Access
> > > We clarify that the requested “Oracle access” is not well-defined in this context: The “key information” goes beyond correct facts and also includes useful knowledge distilled from reasoning and tool-use traces (i.e., skills/experience, aka: **procedural memory**), whose optimal form is not directly verifiable even for humans.
> > >
> > > Again, we try our best to approximate an oracle by constructing “key information”: we inject golden outcomes of prior subtasks and LLM-distilled concise workflows into long-context agents (so minimizing retrieval loss). Results are:
> > > | |Shopping|Travel|Search|Math|
> > > |---|---|---|---|---|
> > > |original|0.01|0.00|0.06|0.26|
> > > | $\Delta$ |+0.016|+0.050|+0.54|+0.16|
> > >
> > > We observe consistent improvements under this simulated oracle. However, overall success rates remain low, suggesting more challenges such as model reasoning over questions and memory.
> > > ## Detailed L0-L4 Results
> > > We show the full L0–L4 results on Travel as requested with long-context agents (i.e., no retrieval or observation loss):
> > > | |L0|L1|L2|L3|L4|
> > > |---|---|---|---|---|---|
> > > |long-context|0.48|0.29|0.21|0.18|0.15|
> > >
> > > Each value is a conditional success rate given previous correct constraints. Since all prior information is fully available in this setting, the monotonic decline cannot be attributed to observation. Instead, it reflects increasing difficulty in reasoning over questions and accumulated memory, especially when memory gets longer (later levels).
> > > ## More results
> > > We provide the requested comparison across all four environments:
> > > | |Shopping|Travel|Search|Math|
> > > |---|---|---|---|---|
> > > |Long-Context(GPT-5.1-mini)|+0.007|+0.000|+0.007|+0.006|
> > > |Memory(Mirix)|+0.008|+0.000|+0.007|+0.001|
> > > |RAG(BM25)|+0.010|+0.000|+0.010|+0.008|
> > >
> > > All differences are consistently small (≤0.01). Even after removing error propagation, performance remains low, suggesting the bottleneck also lies in reasoning over complex queries and long memory, rather than cascading errors alone.
> > >
> > > We hope we addressed your concerns!

---

### Official Review · Reviewer_JwWp · 2026-03-13

**Soundness:** 3
**Presentation:** 3
**Significance:** 4
**Originality:** 3
**Overall Recommendation:** 5
**Confidence:** 4

**Summary:**

The paper argues that existing memory benchmarks mostly evaluate recall in isolation, while existing agent benchmarks usually do not require persistent cross-session memory. To address this gap, it introduces MEMORYARENA, a benchmark of 766 multi-session tasks across bundled web shopping, group travel planning, progressive web search, and formal reasoning. The benchmark is designed so that later subtasks depend on information acquired in earlier sessions, and the paper evaluates long-context agents, RAG variants, and external-memory systems under this setting. The main empirical claim is that systems that perform strongly on prior memory benchmarks still achieve low success on these interdependent agent tasks.

**Compliance With Llm Reviewing Policy:**

Affirmed.

**Final Justification:**

I adjusted score.

**Key Questions For Authors:**

1. Can you provide stronger evidence for the proposed explanations of external-memory underperformance, especially the claimed representation mismatch and training mismatch?

2. How robust are the main conclusions to the choice of task agent? Since the main memory-system comparison fixes GPT-5.1-mini, would the same qualitative ranking and conclusions hold for at least one substantially different task agent?

**Limitations:**

yes

**Strengths And Weaknesses:**

Strengths

The paper addresses a real evaluation gap. Its central point, that useful agent memory should be assessed within a full interaction loop rather than as standalone retrieval, is well motivated, and the benchmark construction broadly matches that motivation. The four environments cover distinct dependency structures, which makes the benchmark more compelling than a single-domain setup.

The work is also potentially significant. The results suggest that strong performance on current memory benchmarks does not straightforwardly transfer to long-horizon, multi-session agent execution. That is a useful empirical message for the community, especially given current interest in memory-augmented agents.

The empirical study is reasonably broad at a systems level, comparing long-context, RAG, and external-memory approaches under a unified setup. The use of progress-style metrics in addition to strict success is sensible, since many tasks appear too difficult for all-or-nothing evaluation alone.

Weaknesses

Some explanatory claims are also insufficiently validated. The discussion of representation mismatch and training mismatch as explanations for why external memory may underperform long-context baselines is plausible, but these are post hoc interpretations rather than results established by targeted ablations.

Presentation needs some improvement. The overall idea is understandable, but the paper is unevenly written and some benchmark details are underspecified. In particular, the related-work positioning on memory benchmarks appears incomplete, for example, BABILong benchmark is not mentioned, which is a notable omission for a paper framing itself partly around limitations of existing long-context / memory evaluation.

Figure 3 would benefit from explicit task-depth legends on the x-axis.


The paper formally defines PS with an equation, but soft Progress Score (sPS) appears only in prose. The lack of a precise formal definition is a reproducibility issue rather than a minor editorial point.

---

> ### Author Rebuttal · Authors · 2026-03-31
>
> Thank you for your insightful comments. We address your concerns below.
>
> **W1:**
> We would like to clarify that the primary contribution of this work is the benchmark itself which introduces new, high-quality evaluation data that exposes previously unobserved behaviors of memory systems in multi-session agentic tasks. Our goal is to provide careful empirical findings and plausible explanations to guide future investigation, rather than to establish definitive causal hypotheses first and then rigorous proof (as is typically the focus of dedicated analysis papers). This is also particularly challenging given the diversity of memory designs and training paradigms.
>
> Encouragingly, the Reviewer HuFN also noted that our *mismatch discussion is a solid explanation*. Moreover, similar phenomena have been observed in prior work (e.g., SkillRL), where even well-trained memory (procedural memory here, also aka *skills*) remains suboptimal compared to jointly optimized with the policy with memory. Together, these provide supporting evidence for our interpretation while highlighting the need for deeper analysis in future work.
>
> **W2 & W3 (Presentation)**
> Thank you for your suggestion in improving paper presentation. We will add the missing reference, BABILong benchmark in the Related Work section. We will also revise the x-axis into Figure 3 according to the suggestion (e.g., add “task-depth at k-th subtask”).
>
> **W4:**  Here is the formal definition of sPS: Formally, consider a test set of $N$ tasks ($\{S_1, S_2, \cdots, S_N\})$ where each task consists of $|S_i|$ ordered subtask ($S_i = [s_1, s_2, \cdots, s_{|S_i|}]$).  For the travel domain, each subtask $s_j$  consists of a set of constraints $C_j =[c_{j_1}, \dots, c_{j_{|C_j|}}]$. Let $|C_j^\text{pass}|$ denote the number of satisfied constraints in $s_j$. We define the soft Progress Score for a task $S_j$ as:
>
> $\text{sPS}_{S_i}$
> $= \frac{1}{|S_i|} \sum _{j=1}^{|S_i|}\frac{|C_j^\text{pass}|}{|C_j|}, $
>
> $\text{sPS} = \frac{1}{N}\sum_{i=1}^N \text{sPS}_{S_i}$
>
> Similar to the hard PS score that each subtask has a binary pass, soft Progress Score (sPS) measures partial satisfaction. This is a continuous generalization of the same notion of progress.
>
> **Reproducibility:**  we promise to fully release MemoryArena dataset, full executable codebase and evaluation scripts, to guarantee transparency and reproducibility.
>
> We will add the above into the final version.
>
> **Q1:**
> We thank the reviewer for this insightful question. Controlled ablations are challenging here due to heterogeneous memory mechanisms across memory systems and some systems are not open-sourced. However, we wanted to note that similar external-memory underperformance has also been observed in prior work (e.g., LoCoMo, Mem0). More discussions about mismatch is also provided in our response to **W1**.
>
> We wanted to clarify that our primary contribution is to provide a high-quality benchmark and careful empirical findings, with plausible explanations to guide future, more targeted analysis.
>
> **Q2 (Another task agent)**
> Following prior work (MemoryAgentBench), we fix a strong task agent to evaluate memory systems, so as to reduce confounding from base model reasoning ability.
> To fully address your concern, we conducted additional experiments with another strong task agent (Claude Sonnet 4.6) across multiple memory systems. The results are listed here:
>
> |  | Shopping SR | Shopping PS | Travel SR | Travel PS | Travel sPS | Search SR | Search PS | Math SR | Math PS | Phys SR | Phys PS |
> |---|---|---|---|---|---|---|---|---|---|---|---|
> | Claude-Sonnet-4.6 | 0.13 | 0.79 | 0.00 | 0.20 | 0.89 | 0.06 | 0.05 | 0.37 | 0.4 | 0.4 | 0.53 |
> | Letta | 0 | 0.48 | 0.00 | 0.00 | 0.15 | 0.23 | 0.11 | 0.16 | 0.27 | 0.4 | 0.53 |
> | Mem0-g | 0 | 0.49 | 0.00 | 0.00 | 0.12 | 0.21 | 0.10 | 0.17 | 0.3 | 0.15 | 0.41 |
> | Text-Embedding-3-Small | 0.04 | 0.55 | 0.00 | 0.04 | 0.42 | 0.27 | 0.09 | 0.37 | 0.35 | 0.63 | 0.68 |
> | MemoRAG | 0.01 | 0.54 | 0.00 | 0.04 | 0.47 | 0.35 | 0.22 | 0.30 | 0.34 | 0.5 | 0.60 |
>
> **The results show consistent trends:** Although some absolute scores are slightly higher (due to a stronger base model), the overall task success rate remains low, and the relative comparison between long-context, memory systems, and RAG is unchanged. This suggests that our findings are robust across task agents.
>
> Notably, we also want to clarify that such evaluations (changing task agent) are computationally expensive: re-running full multi-session experiments (4,850 subtasks, and with 10 memory systems choices) requires substantial API costs. While it is not feasible to exhaustively evaluate as many task agents as possible for a research lab like us, we design our codebase to be modular and easily extensible by replacing task agents or adding new memory systems. We promise to fully release it to enable future studies and customized evaluation.
>
> We will incorporate the above discussion to further strengthen the final version.

---

> > ### Author Rebuttal · Reviewer_JwWp · 2026-04-03
> >
> > I'm satisfied with authors answers and proceed with Accept score.

---

> > > ### Author Response · Authors · 2026-04-08
> > >
> > > We are pleased to hear that our responses addressed your questions! Thank you for your constructive feedback, which helps further strengthen the paper. We will carefully incorporate it into the final version.

---

### Official Review · Reviewer_HuFN · 2026-03-13

**Soundness:** 3
**Presentation:** 3
**Significance:** 3
**Originality:** 3
**Overall Recommendation:** 4
**Confidence:** 3

**Summary:**

This paper introduces MEMORYARENA, a benchmark that tests whether an agent can use memory to make better decisions later. The tasks are interdependent and span multiple sessions in a loop between memory, the agent, and the environment. The benchmark includes bundled web shopping, group travel planning with preferences, progressive web search, and sequential formal reasoning in math and physics. The experiments compare long context baselines with external memory systems and RAG style retrieval, reporting SR and PS plus sPS for group travel and latency. Even strong long memory methods still struggle in this setting.

**Compliance With Llm Reviewing Policy:**

Affirmed.

**Key Questions For Authors:**

1. How do you recommend comparing results across environments when the definitions of success and progress differ.
2. How consistent is the conclusion that external memory or RAG is not always better than long context when you change the base agent to be stronger or weaker.
3. Can you provide a deeper failure breakdown, such as retrieval hit rate, memory update quality, and where errors accumulate across sessions.

**Limitations:**

Main limitations are dataset scale and coverage and limited analysis that separates retrieval errors, memory update errors, compression losses, and downstream integration errors.

**Strengths And Weaknesses:**

Strengths:
1. Clear motivation and a good problem setup. It treats memory as something that should help future actions, not just recall.
2. The tasks are designed so earlier information directly matters for later subtasks, so the benchmark really stresses memory use.
3. The results are useful and a bit surprising. External memory or RAG is not always better than simply keeping the full context, and the mismatch discussion is a solid explanation.

Weaknesses:
- The main contribution is the benchmark rather than a new memory method, so the paper should argue harder for standardization and long term impact.
- Some parts are small and very human crafted, such as 150 shopping tasks, 270 travel tasks, and 40 math plus 20 physics problems, which may limit coverage and scalability.
- Evaluation is not fully consistent across environments, since group travel needs sPS, and many main experiments fix the task agent to GPT 5.1 mini, so it is hard to judge robustness across base models.

---

> ### Author Rebuttal · Authors · 2026-03-31
>
> **W1:**
> We agree and will clarify significance more clearly. MemoryArena introduces multi-session, interdependent tasks where memory must causally support future actions, exposing limitations that existing static or single-session benchmarks miss, making it highly relevant for agentic applications.
>
> Moreover, MemoryArena reflects substantial effort and quality: it involves expert-level annotation (e.g., 8-10h per math/physics task by senior PhDs) and ~$60K for construction and evaluation. We believe this investment highlights both the difficulty and the long-term value of our work.
>
> We will emphasize its standardization and impact more strongly in the final version.
>
> **W2:**
> 1. **Scale**: MemoryArena is comparable in scale to agentic benchmarks (ToolAthlon:108, MCP-Bench:104, WebArena:821). Due to compositional multi-session design, it expands to **4,850** subtasks (same task granularity to e.g. WebShop). Math/phys contains **440 subtasks**, yielding larger effective scale and lower sensitivity to randomness.
> 2. **Coverage**: Math/phys spans diverse domains (e.g., pure math, high-energy physics, condensed matter...) at research-paper-level difficulty rather than textbook in previous datasets. This quality reflects a deliberate trade-off toward high-quality data resources over superficial scale.
>
> **W3**
> 1. **PS/sPS**
> The objective is to measure the progress (extent of completion) of multi-session tasks, which can be defined either in a binary (hard) form or a non-binary (soft) form. In Travel, each subtask has ~8 coupled constraints over long dependency, making binary subtask completion almost always 0 and thus uninformative.
> To provide more meaningful insights, we report a soft Progress Score (sPS) that measures partial satisfaction. This is a continuous version of progress score, rather than a different metric.
> 2. **New results w/ Claude Sonnet 4.6**
> Following prior work (MemoryAgentBench), we fix a strong task agent to evaluate memory systems. However, to address the concern, we provided additional results with a stronger task agent (Claude Sonnet 4.6):
> |  | Shopping SR | Shopping PS | Travel SR | Travel PS | Travel sPS | Search SR | Search PS | Math SR | Math PS | Phys SR | Phys PS |
> |---|---|---|---|---|---|---|---|---|---|---|---|
> | Claude-Sonnet-4.6 | 0.13 | 0.79 | 0.00 | 0.20 | 0.89 | 0.06 | 0.05 | 0.37 | 0.4 | 0.4 | 0.53 |
> | Letta | 0 | 0.48 | 0.00 | 0.00 | 0.15 | 0.23 | 0.11 | 0.16 | 0.27 | 0.4 | 0.53 |
> | Mem0-g | 0 | 0.49 | 0.00 | 0.00 | 0.12 | 0.21 | 0.10 | 0.17 | 0.3 | 0.15 | 0.41 |
> | Text-Embedding-3-Small | 0.04 | 0.55 | 0.00 | 0.04 | 0.42 | 0.27 | 0.09 | 0.37 | 0.35 | 0.63 | 0.68 |
> | MemoRAG | 0.01 | 0.54 | 0.00 | 0.04 | 0.47 | 0.35 | 0.22 | 0.30 | 0.34 | 0.5 | 0.60 |
>
> **Results show consistent trends:**: although absolute scores improve slightly (due to a stronger base model), the task success rate remains low, and the relative comparison between long-context, memory systems, and RAG is unchanged. This suggests that our findings are robust across task agents.
>
> Notably, re-running full evaluation (4,850 subtasks across 10 memory systems) is computationally expensive and incurs substantial API cost, making exhaustive evaluation over many task agents impractical. We design our codebase to be modular and easily extensible by replacing task agents or adding new memory. We will fully release our codes for customized evaluation.
>
> **Q1**:
> SR measures whether a task is fully completed, while PS measures how much of the task is completed via subtask success. As explained in our response to W3, sPS is simply a continuous relaxation of PS, measuring the same thing.
>
> Thus, these metrics reflect the same underlying objective -- task completion, from complementary perspectives (strict vs. partial). This makes results across environments interpretable in a consistent way, even when task structures differ.
>
> **Q2**:
> As shown in our response to W3, memory results with a stronger task agent (Claude Sonnet 4.6) exhibit the same qualitative trends. Comparing with a weaker agent (GPT-5.1-mini), we observe that when memory/RAG provides little benefit, a stronger base model further widens the gap in favor of the no-memory (long-context) setting. Conversely, when memory/RAG offers some gains, those gains reduce as the base agent becomes stronger.
>
> **Q3**:
> For some memory systems (e.g., Mirix, Letta, Mem0-g), their retrieval and memory update processes are not accessible, making it difficult to measure retrieval hit rate or assess update quality.
>
> Error accumulation: we perform a controlled diagnostic by injecting ground-truth subtask outcomes into memory (removing dependence on prior errors). This yields no significant improvement (e.g., Shopping w/ Mirix: 0.008 diff), suggesting cascading errors are not the primary driver of failure. Consistently, methods designed for independent tasks (e.g., procedural memory in ReasoningBank, no error propagation) also perform poorly on MemoryArena, supporting this conclusion.

---

### Official Review · Reviewer_SaGB · 2026-03-14

**Soundness:** 2
**Presentation:** 3
**Significance:** 3
**Originality:** 3
**Overall Recommendation:** 4
**Confidence:** 4

**Summary:**

The paper introduces MemoryArena, a gym designed to evaluate the functional utility of memory in LLM agents through multi-session, interdependent tasks. Unlike benchmarks that test passive recall, MemoryArena requires agents to distill experience from earlier interactions to satisfy specific constraints and preferences in future sessions. By using human-crafted scenarios with explicitly linked subtasks, it reveals critical performance gaps between long-context models and modular RAG-based systems, which often fail due to retrieval bottlenecks.

**Compliance With Llm Reviewing Policy:**

Affirmed.

**Final Justification:**

My concerns have been fully addressed by authors' rebuttals and I will recommend this paper as acceptance.

**Key Questions For Authors:**

- Does the benchmark include a memory budget (token limit)? How do models perform when forced to use a summarization module versus raw log retrieval?
- If a user provides a correction in Session 3 that invalidates a fact from Session 1, how is the agent's internal knowledge change and how this could be measured?
- In your RAG baseline, if you replaced BM25 with a state-of-the-art dense vector retriever, does the gap between RAG and Long-Context models (Figure 5) diminish significantly?
- Do any tasks require remembering implicit user intent (unspoken preferences derived from history) rather than explicit keyword-based constraints?

**Limitations:**

- The gym focuses on a stable world where only the agent's history changes. It does not model world-state decay or external events that happen between sessions, which would test the agent's ability to distinguish between stale and fresh memory.
- The benchmark is limited to a single-agent, single-user scenario. It does not address the challenges of multi-agent memory sharing or privacy-preserving memory boundaries.

This paper proposes the cross-session agent memory benchmarks, which are promising. I would be happy to raise my score if the authors can help address the concerns.

**Strengths And Weaknesses:**

### Strengths
- The motivation is good. It moves beyond memorization as recall to memory as a decision prior, which is more representative of real-world agentic deployment.
- The use of negative constraints provides a clear binary for success, reducing the ambiguity often found in LLM-evals.
- It provides human-crafted interdependencies, which are important to the field of agent memories to be more realistic.

### Weaknesses
- In Figure 5, the failure of the GPT-5-mini + BM25 setup is attributed to memory, but it appears to be a retrieval-stage failure rather than a reasoning or memorization failure. The benchmark may be measuring the brittleness of standard retrieval algorithms more than the agent’s memory management.
- The benchmark assumes the agent's policy is fixed during the loop. It does not account for online policy adaptation where the agent might change its data-saving strategy based on environment feedback.
- As a human-crafted benchmark, it lacks the massive scale of procedurally generated datasets, potentially limiting its use as a training set for fine-tuning memory-native models.

---

> ### Author Rebuttal · Authors · 2026-03-31
>
> **W1:**
> Thanks for the insightful comment.  We would like to clarify:
>
> 1. **Retrieval is an integral part of memory management:** A functional memory system requires both memory construction and memory retrieval. If relevant information cannot be retrieved, it is effectively unusable. Thus, retrieval failures directly reflect memory failures rather than an orthogonal issue.
> 2. **MemoryArena goes beyond standard retrieval:** We evaluate diverse memory systems (e.g., Letta, Mirix, Mem0(-G), GraphRAG; Table 3) that involve abstraction, consolidation, and structured memory. Moreover, long-context agents (without any retrieval module)  also exhibit similarly low performance. This suggests the limitation is not specific to retrieval, but reflects a more fundamental challenge in memory management across sessions.
> 3. **Appx. Fig 5 is an illustrative example, not primary evidence:** Fig 5 shows one representative failure case for interpretability. Our conclusions are based on aggregate results across tasks rather than a single example (e.g., Fig 4 demonstrates a different failure mode unrelated to retrieval).
> 4. **End-to-end evaluation of memory with new data:** Unlike prior benchmarks that isolate recall, MemoryArena evaluates memory as a functional component in multi-session agent behavior, where retrieval, reasoning, and action are inherently coupled. Our data also provides high-quality resources for the community.
>
> **W2&L1(online vs. stale)** MemoryArena evaluates online, evolving memory within the agent loop: the task agent conditions on prior memory which directly alters its behavior (similar findings are in ReasoningBank paper, etc) , while memory construction itself is adaptive to prior interactions and environment feedback. This creates a closed loop, showing continuous online adaptation in task agent and memory agent without parameter updates.
>
> Importantly, tasks are intentionally designed such that past information remains relevant for future decisions.so memory is continuously updated and reused rather than becoming “stale”.
>
> **W3:**
> **MemoryArena is already comparable in scale to existing agentic benchmarks** (e.g., ToolAthlon: 108; MCP-Bench: 104; WebArena: 821).  Moreover, each of our tasks is compositional and multi-session. When counted at the subtask level (comparable to task granularity in e.g., WebArena/Webshop), this corresponds to **4,850 agentic subtasks**, indicating a substantially larger effective scale.
>
> Finally, MemoryArena is designed as an evaluation benchmark, not a training dataset, similar to most prior benchmarks (e.g., WebArena). Its human-crafted design ensures high-quality, challenging tasks that are difficult to obtain via procedural generation.
>
> **Q1:**
> Imposing a fixed memory budget is challenging due to the heterogeneous designs of memory systems (often with black-box components), and may not reflect realistic usage.
> Importantly, MemoryArena already evaluates systems with summarization and consolidation mechanisms rather than just raw log retrieval. More details in response to W1 (the 2nd point).
>
> **Q2:** Many memory agents support memory updating mechanisms (e.g., Mirix has ‘episodic(semantic)_memory_update’ tools, Letta has ‘memory_update’), allowing stored information to be revised with new evidence, whereas standard RAG typically lack this capability.
>
> Evaluating knowledge conflict resolution has been studied extensively in prior QA benchmarks [MemoryAgentBench, LongMemEval]. MemoryArena instead focuses on downstream multi-session performance, where successful task completion implicitly requires resolving such conflicts.
>
> **Q3:** Yes. We already include a dense retrieval baseline (text-embedding-3-small) in Table 3, which improves over BM25. However, the gap with long-context models persists.
>
> **Q4, L1(stable world), L2:** Thank you for these suggestions. However, each introduces additional challenges beyond memory: modeling implicit user intent is still under exploration in user modeling and alignment [1,2]; adding external world events requires handling exogenous variables [3]; multi-agent settings introduce additional complexities that are not yet well standardized [4]. Thus, these are orthogonal to the current goal of MemoryArena.
>
> Moreover, extending to these settings would further increase task complexity. But even under current setup, we already observe substantial performance gaps, showing that memory utilization alone is already highly challenging and provides new insights beyond prior benchmarks.
>
> We agree these valuable extensions and are happy to explore them in future work. We will add these limitation discussions into our final paper.
>
> 1. DISCOVERLLM: From Executing Intents to Discovering Them
> 2. Aligning LLMs with Individual Preferences via Interaction, COLING25
> 3. Discovering and Removing Exogenous State Variables and Rewards for Reinforcement Learning, ICML18
> 4. Multi-Agent Memory from a Computer Architecture Perspective: Visions and Challenges Ahead

---

> > ### Author Rebuttal · Reviewer_SaGB · 2026-04-06
> >
> > My concerns have been fully addressed and I will raise my score to 4.

---

> > > ### Author Response · Authors · 2026-04-08
> > >
> > > We are delighted to hear that your concerns have been fully addressed. Thank you for your insightful reviews and constructive discussion! We will carefully incorporate your suggestions into the final version.

---

### Decision · Program_Chairs · 2026-04-30

**Decision:**

Accept (regular)

**Comment:**

This paper introduces MEMORYARENA, a benchmark for evaluating agent memory within multi-session agentic loops where memorization and action are tightly coupled. The reviewers highlight the benchmark's design for addressing an important evaluation gap: assessing memory as a decision prior within full interaction loops rather than standalone retrieval. The four environments (shopping, travel, math, physics) cover distinct dependency structures, and the use of negative constraints provides clear success criteria. During the rebuttal, the authors provided additional experiments with Claude Sonnet 4.6 showing consistent trends across task agents, a formal definition of sPS, and a diagnostic experiment injecting ground-truth outcomes to rule out error cascading as the primary failure driver. Two reviewers explicitly confirmed full resolution of concerns (SaGB raised score to 4; JwWp confirmed Accept). Reviewer ZqnJ's remaining concern about dataset size in some domains (40 math, 20 physics) is acknowledged but does not undermine the benchmark's core contribution. Minor residual concerns include the post-hoc nature of the representation/training mismatch hypotheses and the lack of formal sPS definition in the original submission (now addressed). These are addressable in the camera-ready version.